# The Cooper-Pair Distribution Function of Untwisted-Misaligned Bilayer Graphene

**DOI:** 10.3390/ijms252312549

**Published:** 2024-11-22

**Authors:** Jose Alfredo Camargo-Martínez, Guillermo Iván González-Pedreros, Fredy Mesa

**Affiliations:** 1Grupo de Investigación en Ciencias Básicas, Aplicación e Innovación CIBAIN, Unitrópico, Yopal 850002, Colombia; jcamargo@unitropico.edu.co; 2Grupo de Ciencias e Ingeniería CEI, Facultad de Ciencias, Universidad Pedagógica y Tecnológica de Colombia, Tunja 150003, Colombia; guillermo.gonzalez02@uptc.edu.co; 3Grupo NanoTech, Facultad de Ingeniería y Ciencias Básicas, Fundación Universitaria Los Libertadores, Bogotá 111221, Colombia

**Keywords:** Cooper-pairs, graphene, electric field, magic angle

## Abstract

The Cooper-pair distribution function $D_{cp}\left(\omega ,T_{c}\right)$ of Untwisted-Misaligned Bilayer Graphene (UMBLG) in the presence of an external electric field is calculated and analysed within the framework of first-principle calculations. A bilayer graphene structure is proposed using a structural geometric approximation, enabling the simulation of a structure rotated at a small angle, avoiding a supercell calculation. The $D_{cp}\left(\omega ,T_{c}\right)$ function of UMBLG indicates the presence of the superconducting state for specific structural configurations, which is consistent with the superconductivity in Twisted Bilayer Graphene (TBLG) reported in the literature. The $D_{cp}\left(\omega ,T_{c}\right)$ function of UMBLG suggests that Cooper-pairs are possible in the low-frequency vibration region. Furthermore, the structural geometric approximation allowed the evaluation of the effect of the electric field on the superconducting state of UMBLG and its superconducting critical temperature through the $N_{cp}$ parameter.

## 1. Introduction

Graphene [1,2] and its derivatives have become a bridge between two seemingly distant fields: molecular biophysics and condensed matter physics. These materials have attracted significant attention in areas such as molecular biology and molecular medicine due to their exceptional thermal, mechanical, optical, and electronic properties, including high electron mobility, zero effective mass, transparency to the visible spectrum, high strength, ultra-fast light response, and zero-bandgap energy [3,4,5,6]. Recent advancements in functional graphene composites have demonstrated excellent biomolecule stability and a large surface area for molecule immobilization. These properties make them highly suitable for developing more efficient chemical, biological, and electronic sensors [7,8,9,10,11,12]. This highlights the importance of approaches aimed at understanding the behavior of graphene-based systems.

Twisted Bilayer Graphene (TBLG) exposed to a constant electric field undergoes a zero-resistance state at a temperature below 1.7 K [13]. The TBLG has remarkable properties: superconductivity [13], Mott insulator behavior [14], and suppression of group velocity at certain angles and its relationship with electronic correlation [15,16,17,18]. The relative twisted angle between layers, in 2D materials such as TBLG [19,20,21], has become an undeniable phenomenon and a new field. In this way, the magic angle value provided a novel knob for tuning superconducting properties [13] and many-body effects [14,22,23]. By contrast, recently superconductivity was reported in non-twisted rhombohedral trilayer graphene, a system without Moiré patterns [24]. This could lead one to think that a specific piece of a TBLG could originate the superconductor state. Nevertheless, to achieve this specific piece, from a technical point of view, it is easier to twist a set of layers than to achieve very tiny displacements between them. To a first approximation in this structure, it can be assumed that the primary mechanism leading to the formation of Cooper-pairs, among other possible candidates, is the electron–phonon interaction. For superconductivity to occur, the electrons in a Cooper-pair and their interaction via phonons must satisfy energy- and momentum-conservation laws. Additionally, this interaction must provide sufficient energy to overcome the repulsive Coulomb interaction, allowing the superconducting phase to become viable. Each of these conditions corresponds to a probability. If then a sum of the overall electronic energies is performed over the product of these probabilities, a simultaneous probability function is obtained to obtain the Cooper-pairs as a function of lattice vibrational energy and $T_{c}$, namely the $D_{cp}\left(\omega ,T_{c}\right)$ function [25]. This function is built from the well-established Eliashberg spectral function and the phonon density of states, calculated by first principles (for more details, see reference [25]). The Cooper-pair distribution function $D_{cp}\left(\omega ,T_{c}\right)$ promises to provide information about the superconductor state through the determination of the spectral regions of Cooper-pairs formation. In addition, from the $D_{cp}\left(\omega ,T_{c}\right)$ function, it is possible to obtain the $N_{cp}$ parameter, which is proportional to the total number of Cooper-pairs formed at temperature $T_{c}$ [26,27]. The $D_{cp}\left(\omega ,T_{c}\right)$ function has been used to evaluate some superconductor systems [25,26,27,28].

In this paper, we aim to contribute to the understanding of superconduction fundamentals of challenging structures such as TBLG. We study an untwisted-misaligned bilayer graphene (UMBLG) structure, which mimics a specific region of a TBLG lattice, using the Cooper-pair distribution function $D_{cp}\left(\omega ,T_{c}\right)$. Here, we report and analyze the $D_{cp}\left(\omega ,T_{c}\right)$ function of UMBLG structures in the presence of a perpendicular external electric field, at temperature T=0.5 K. The findings show that superconductivity in these structures responds to slight relative displacement between graphene sheets and to external electric fields, as in magic angle TBLG.

## 2. Results and Discussion

The relaxation procedure leads to a lattice constant a=2.47 Å, which is in good agreement with other theoretical reports [29,30]. On the other hand, we find out the inter-layer distance d=4.40 Å, By contrast, experimental and theoretical reports are 3.35 Å and 3.31 Å [29,31] for AB-stacked BLG. The discrepancy is due to the fact that the bilayer has been isolated. To do so, we set the AB-stacked bilayer up into a cell with a vertical length of more than eleven times the lattice constant to dismiss the inter-bilayer interaction. We used the Quantum Espresso code [32] for all these calculations.

Figure 1 shows the Eliashberg function $\alpha^{2}F\left(\omega \right)$ and Phonon density of states (PhDOS) calculated for the AB-stacked bilayer graphene proposed in the presence of a perpendicular electric field proportional to $E_{0}=9.0$ mV/Å, and AB-stacked untwisted-misaligned bilayer graphene (associated with $\theta_{33}=0.987{}^{\circ}$ and $\theta_{32}=1.018{}^{\circ}$ angles of a TBLG) in the presence of a perpendicular electric field of $5E_{0}$. These spectra show four characteristic peaks located approximately at 60, 80, 170 and 200 meV, which are consistent with the theoretical calculations of AB-stacked bilayer graphene (with 10.000 atoms) reported by Choi et al. [33,34] using $\theta_{m}=1.08{}^{\circ}$, and PhDOS of TBLG at $\theta_{m}=13{}^{\circ}$ and 21° (with 28 and 76 atoms, respectively) reported by Cocemasov et al. [35].

It can be observed from Figure 1 that the $\alpha^{2}F\left(\omega \right)$ and PhDOS are nearly insensitive to the small displacements associated with twist angle ($\theta_{m}$) as well as to the presence of an electrical field. In addition, the insensitivity of PhDOS (TBLG) to the twist angle was observed by Cocemasov et al. [35]. Therefore, it is almost not possible to establish significant differences induced by displacements (associated with $\theta_{m}$) or $E_{0}$, more than those observed below 5 meV (see inset of Figure 1). At first sight, these slight differences do not seem to offer significant information about variations in the UMBLG properties. However, with these spectra, it is possible to obtain the Cooper-pair distribution function $D_{cp}\left(\omega ,T_{c}\right)$, which allows evaluation of the system in the superconducting state [25,27].

Figure 2 shows the $D_{cp}\left(\omega ,T_{c}\right)$ function of AB-stacked untwisted-misaligned bilayer graphene (associated with angles $\theta_{m}=0{}^{\circ},0.987{}^{\circ}$ and 1.018°) in the presence of an electrical field. The $D_{cp}\left(\omega ,T_{c}\right)$ function of UMBLG suggests that Cooper-pairs are possible in the low-frequency vibration region. By contrast, BLG with a perpendicular electric field of magnitude $E_{0}$ (Figure 2a) is in the normal state (non-superconducting). It is inferred because it reports a $D_{cp}\left(\omega ,T_{c}\right)\approx 0$. Namely, the superconducting phase is theoretically observing if the $D_{cp}\left(\omega ,T_{c}\right)$ function is not zero, which, in this case, is achieved when bilayer graphene is displaced (associated with a small rotation angle) in the presence of the electric field. This condition matches the experimental reports (superconductivity in magic-angle graphene superlattices [13]). This suggests the origin of superconductivity in this kind of structure could be a slightly misaligned piece. It is also evident from Figure 2 that the $D_{cp}\left(\omega ,T_{c}\right)$ function reveals a significant sensitivity to displacements associated with small variations of $\theta_{m}$ as well as to the presence of an electric field (note the scale differences between Figure 2a and Figure 2b). This is possibly due to the correlational analysis of the electronic and phononic properties of the system that the $D_{cp}\left(\omega ,T_{c}\right)$ function achieves, which cannot be determined from an individual evaluation of such properties, as was evidenced with the results presented in Figure 1. However, the sensitivity of the $D_{cp}\left(\omega ,T_{c}\right)$ function is a criterion that requires more calculations to be validated, in addition to those already carried out (H_3_S [25], Nb [26] and D_3_S [27]).

The $D_{cp}\left(\omega ,T_{c}\right)$ function of MUBLG shows a quasi-Gaussian form, which is appreciably affected by $\theta_{m}$ and the presence of an electric field. It is observed that an increase in the intensity of the electric field leads to an increase in the area under the spectral line and moves it slightly towards higher values of frequencies, for both angles. However, the effect was more marked for $\theta_{m}=0.987{}^{\circ}$ (note the difference in the scale of the vertical axes in Figure 2). $D_{cp}\left(\omega ,T_{c}\right)$ is only observed at frequencies below 0.4 meV (centered around 0.05 meV), which seems consistent with the low critical temperature of TBLG (Tc=0.5 K [13]).

Furthermore, from the $D_{cp}\left(\omega ,T_{c}\right)$ function it is possible to obtain an estimate of the total number of Cooper-pairs formed at temperature $T_{c}$ through a quantity proportional to it, the $N_{cp}$ parameter [26,27]. The comparison of the $N_{cp}$ parameters obtained from each $D_{cp}\left(\omega ,T_{c}\right)$ as a function of an electrical field, for both misaligned structures which are associated with the two angles, is shown in Figure 3.

It can be observed in Figure 3 that an electric field has a greater effect on the $N_{cp}$ parameters obtained for UMTBLG associated with the angle θ=0.987°, compared to θ=1.018°. This could be because the electric field affects the flat band physics (dispersion and topological properties) [36,37], which are associated with the superconducting phase. Since a high $N_{cp}$ value implies a high $T_{c}$ [26,27], it can be suggested from Figure 2 that an increase in the intensity of the electric field could lead to an increase in $T_{c}$ in TBLG, for a specific angle of rotation. In addition, calculations for $D_{31}=0.122988$ nm ($\theta_{31}=1.05{}^{\circ}$) were also performed. Outcomes showed the corresponding $D_{cp}\left(\omega ,T_{c}\right)$ function much less than $D_{32}=0.123024$ nm ($\theta_{32}=1.018{}^{\circ}$), so it was not included in Figure 2 and Figure 3.

In comparison to a previous analysis (see Figure 3 in ref. [28]), our results (see Figure 2) reveal that the presence of the electric field induces a shift (∼0.2 meV) towards lower frequencies. Furthermore, when comparing the configuration at θ=1.05°, the electric field causes a decrease in the area under the $D_{cp}$ curve, which, according to the interpretation of the Ncp parameter, implies a reduction in the number of Cooper-pairs and, consequently, a decrease in Tc. It is important to note that no significant changes are observed below 200 meV in the $\alpha^{2}F\left(\omega \right)$ functions due to the electric field, which underscores the scope and resolution of the $D_{cp}\left(\omega ,T_{c}\right)$ function.

Finally, despite the consistent results shown up to now, some expression of the characteristic flat bands of TBLG [36] are not observed directly in the version of $D_{cp}\left(\omega ,T_{c}\right)$ used in this study. This requires further analysis.

## 3. Materials and Methods


In order to build the $D_{cp}\left(\omega ,T_{c}\right)$ function, calculations of electronic density states, vibrational density states, and the Eliashberg function of the concerned structure are required. To perform these ab initio calculations, we first relaxed the internal degrees of freedom and the lattice vectors of the crystal structure using the Broyden–Fletcher–Goldfarb–Shanno (BFGS) quasi-Newton algorithm [38,39]. From these relaxed structure configurations, electronic and phonon band structures, electron (DOS) and phonon (PhDOS) densities of states, and the Eliashberg function $\alpha^{2}F\left(\omega \right)$ are calculated, for this study, in the presence of the perpendicular electric fields $E_{0}$, 3$E_{0}$, and 5$E_{0}$ with $E_{0}=9$ mV/Å. We used a kinetic energy cut-off of 70 Ry for the expansion of the wave function into plane waves and 280 Ry for the density. To integrate over the Brillouin zone, we used for the electronic integration a k-grid of 32 × 32 × 1 and for the phononic integration a q-grid of 8×8×1, according to the Monkhorst–Pack scheme [40]. We performed the calculations using the pseudopotential plane-wave (PW) method of Perdew et al. [41], the generalized gradient approximation (GGA), and a Troullier and Martins [42] norm-conserving pseudopotential. The cut-off and grids were chosen big enough to obtain good precision in $\alpha^{2}F\left(\omega \right)$ calculated within the density-functional perturbation theory (DFPT) frame [30,43]. We used the Quantum Espresso code [32] for all these calculations.

### 3.1. The Cooper-Pair Distribution Function

Conventional superconductivity can be explained by an attractive electron–electron interaction (Cooper-pair) mediated by the lattice phonons. Cooper-pair formation is induced by the simultaneous occurrence of specific physical conditions, which can be described as a product of probabilities in terms of electronic and vibrational states, and electron–phonon interaction of a system summed over all electronic states defines the Cooper-pair distribution function $D_{cp}\left(\omega ,T_{c}\right)$, given by
(1)$$D_{cp}\left(\omega ,T\right)=\int_{E_{F}-\omega_{c}}^{E_{F}+\omega_{c}}\int_{E_{F}-\omega_{c}}^{E_{F}+\omega_{c}}g_{ep}^{s}\left(T,\epsilon ,\omega \right)\times g_{ep}^{b}\left(T,\epsilon^{\prime }+\omega ,\omega \right)\times \alpha^{2}\left(\omega \right)d\epsilon d\epsilon^{\prime },$$
where $g_{ep}^{s}\left(T,\epsilon ,\omega \right)=g_{e}^{o}\left(T,\epsilon \right)g_{e}^{v}\left(T,\epsilon +\omega \right)g_{p}^{n}\left(T,\epsilon \right)$ and $g_{ep}^{b}\left(T,\epsilon ,\omega \right)=g_{e}^{o}\left(T,\epsilon \right)g_{e}^{v}\left(T,\epsilon -\omega \right)g_{p}^{n+1}\left(T,\epsilon \right)$ are the simultaneous probability at *T* related with the presence of electronic states occupied $g_{e}^{o}$ and available $g_{e}^{v}$ (near to the Fermi level), as well as the probabilistic weight of optimal vibrational states $g_{p}^{n}$, which are candidates for the consolidation of an attractive interaction potential that drives Cooper-pair formation. Further, $\alpha^{2}\left(\omega \right)=\alpha^{2}F\left(\omega \right)/N_{ph}\left(\omega \right)$ establishes the electron–phonon coupling probability [44,45] through the well-known Eliashberg spectral function and phonon density of states. In this sense, the $D_{cp}\left(\omega ,T_{c}\right)$ function provides the spectral range where Cooper-pairs could be formed.

Furthermore, from the $D_{cp}\left(\omega ,T_{c}\right)$ function it is possible to estimate the total number of Cooper-pairs formed at the temperature through a quantity proportional to it, the $N_{cp}\left(T\right)$ parameter, given by
(2)$$N_{cp}\left(T\right)=\int_{0}^{\omega_{c}}D_{cp}\left(T,\omega \right)d\omega ,$$
where ω is phonon energy and $\omega_{c}$ is a cutoff energy, so that to $\omega >\omega_{c}$$D_{cp}\left(\omega ,T_{c}\right)$ is negligible.

### 3.2. Structure

Figure 4 illustrates the AB-stacked TBLG structure. The top view, in Figure 4a, shows two hexagonal layers, where one is rotated with respect to the other by a small angle $\theta_{m}$, and Figure 4b illustrates the side view showing the corresponding interlayer distance *d*. Figure 5 summarizes the geometric concepts used to determine the structural, electronic and vibrational properties of the AB-stacked TBLG structure. The rotation angle $\theta_{m}$ in a commensurate rotation condition ($\theta =\theta_{m}$) [46] is determined as follows:(3)$$\cos\left(\theta_{m}\right)=\frac{3m^{2}+3m+1/2}{3m^{2}+3m+1},$$
with m=0,1,2,3,…, which allows a long-wave-length Moiré pattern with $\lambda_{m}=$$a/(2\sin\left(\theta_{m}/2\right))$ [47].

With the purpose of establishing a possible UMBLG structure associated with a TBLG structure, we initially consider a supercell of an AB-stacked BLG rotated at a specific small angle θ (Figure 5b), in which were identified two equivalent regions where AB-stacked TBLG configurations are symmetric particularly around a rotation axis (see Figure 5a). In the midpoint of these regions, corresponding to $\lambda_{m}/2$, we encounter a lattice arrangement akin to misaligned bilayer graphene, which is observed as two AB-stacked layers with a parallel displacement between them, as illustrated in Figure 5c. We established that the value of this displacement $D_{m}$ is linked to the twisted angle $\theta_{m}$ of the associated TBLG structure. Moreover, this displaced structure can be easily built from an untwisted bilayer graphene (Figure 5d). Table 1 shows the geometrical characteristics of the three configurations, and the respective displacement values $D_{m}$ associated with each twisted angle $\theta_{m}$.

When the layers are neither twisted nor misaligned, a superconducting state cannot be achieved. This occurs when specific rotational angles align the layers in such a way that they lack the necessary displacement. In this configuration, regions near the rotation axis or any equivalent point within the superlattice closely resemble the untwisted and aligned case. This suggests that the origin of superconductivity does not lie in these regions but rather in the areas most affected by the rotation—specifically, those near the midpoint of the Moiré superlattice, around the point $\lambda_{m}/2$. The structure proposed in this study consists of four carbon atoms (two per layer) whose positions and internal parameters generate two displaced graphene layers, which reproduce the AB-stacked BLG structure in the zone $\lambda_{m}/2$. In this unit cell, the interlayer distance is set to 4.40 Å, which is sufficient to establish interactions between the layers. A 12 Å vacuum gap is applied perpendicular to the slab surface to eliminate interactions between periodic images, ensuring that the bilayer system is effectively isolated.

## 4. Conclusions

In this study, we reported the Cooper-pair distribution function $D_{cp}\left(\omega ,T_{c}\right)$ of Untwisted-Misaligned Bilayer Graphene (UMBLG), under the influence of external electric fields. The UMBLG structure was proposed as a structural geometric approximation that mimics a characteristic region of a Twisted Bilayer Graphene (TBLG) structure. This UMBLG structure was modeled as two AB-stacked layers with a parallel displacement $D_{m}$ between them. We consider three displacements $D_{33}=0.12295$ nm, $D_{32}=0.12302$ nm, and $D_{31}=0.12299$ nm, which are linked with the rotation angles (near the magic angle) $\theta_{33}=0.987{}^{\circ}$, $\theta_{32}=1.018{}^{\circ}$, and $\theta_{31}=1.05{}^{\circ}$, respectively, associated with Twisted Bilayer Graphene (TBG). The $D_{cp}\left(\omega ,T_{c}\right)$ function of bilayer graphene in the presence of an electric field was also calculated, as a reference. As expected, the reference structure showed a negligible $D_{cp}\left(\omega ,T_{c}\right)$ function due to the fact that it is not a superconductor one. However, the $D_{cp}\left(\omega ,T_{c}\right)$ function of UMBLG structures with $D_{33}$ and $D_{32}$ displacements showed characteristics indicative of superconducting behavior, which became more noticeable with the increase in the electric field. This behavior was greater for $D_{33}$($\theta_{33}=0.987{}^{\circ}$) than for $D_{32}$($\theta_{32}=1.018{}^{\circ}$). Furthermore, the $D_{cp}\left(\omega ,T_{c}\right)$ function of these structures suggests that Cooper-pairs are possible in the low-frequency vibration region. In addition, the calculation of the $N_{cp}$ parameters allowed us to suggest that for specific displacements, an increase in $N_{cp}$ could be induced by applying and increasing the external perpendicular electric field. This implies an increase in the number of Cooper-pairs and therefore an increase in the superconducting critical temperature. The displacement $D_{31}$ in the UMBLG structure indicated that it does not exhibit superconductivity under this particular configuration. All these results demonstrated that the structural geometric approximation appears to be consistent with the presence of the superconducting state observed in TBLG at small angles, which allowed evaluation of the possible effect of the electric field.

## Figures and Tables

**Figure 1 ijms-25-12549-f001:**
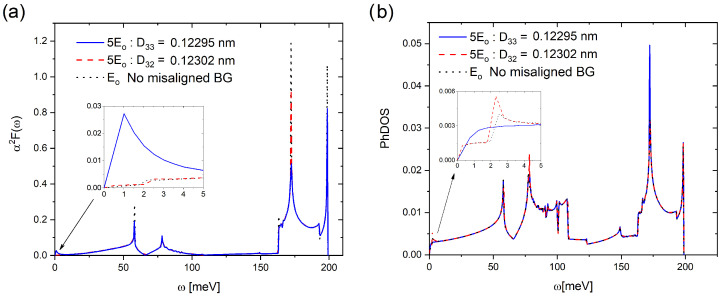
(**a**) Eliashberg function $\alpha^{2}F\left(\omega \right)$ and (**b**) Phonon density of states (PhDOs) of untwisted-misaligned bilayer graphene (UMBLG) at $D_{33}$ and $D_{32}$, in the presence of a perpendicular electric field ($E_{0}=9.0$ mV/Å, 3$E_{0}$ and 5$E_{0}$). Insets show a low-frequency range.

**Figure 2 ijms-25-12549-f002:**
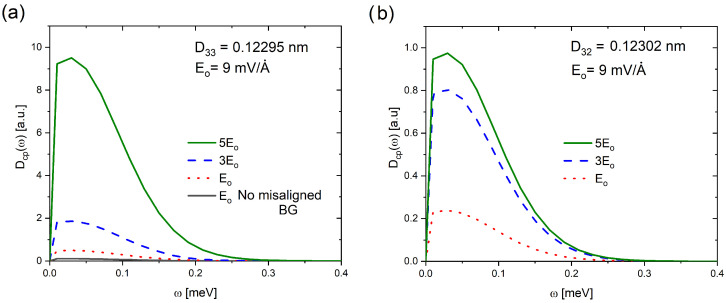
The Cooper-pair distribution function $D_{cp}\left(\omega ,T_{c}\right)$ of untwisted-misaligned bilayer graphene (UMBLG) for (**a**) $D_{33}$ and (**b**) $D_{32}$, in the presence of a perpendicular electrical field. In Figure (**a**), a not-twisted BLG is included. Note the difference in the scale of the vertical axes.

**Figure 3 ijms-25-12549-f003:**
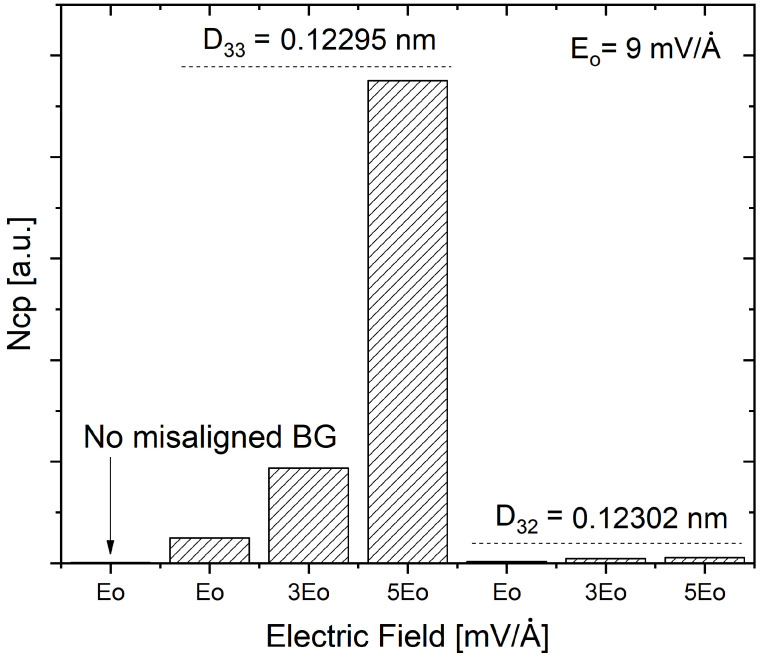
Comparison of the $N_{cp}$ parameters, $N_{cp}=\int_{0}^{\omega_{c}}D_{cp}\left(\omega ,T_{c}\right)d\omega$, as a function the electric field $E_{0}$, of untwisted-misaligned bilayer graphene (UMBLG) for $D_{33}$ and $D_{32}$, and no misaligned bilayer graphene.

**Figure 4 ijms-25-12549-f004:**
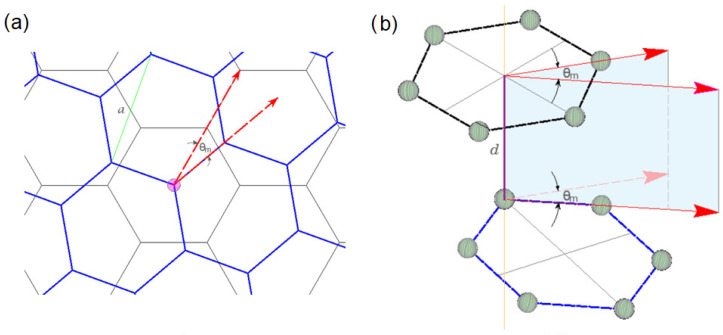
TBLG structure: (**a**) top view of two hexagonal layers in AB configuration with lattice constant a rotated angle $\theta_{m}$; (**b**) side view of two twisted hexagonal layers in AB configuration, separated distance *d*.

**Figure 5 ijms-25-12549-f005:**
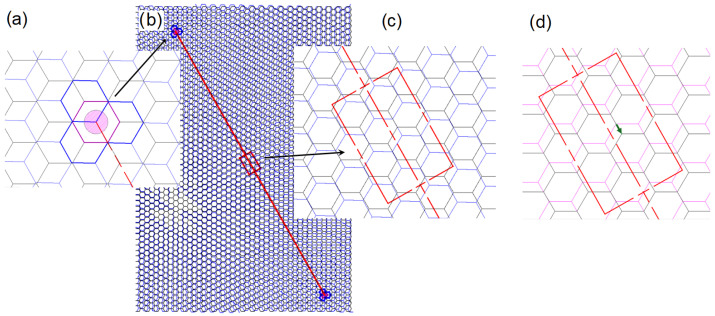
(**a**) Two layers rotated at small angle $\theta_{m}$ around an axis; (**b**) supercell of an AB-stacked TBLG; the red line represents a long-wavelength $\lambda_{m}$; (**c**) inset of the region (red rectangle) in $\lambda_{m}/2$; (**d**) two hexagonal layers displaced between them (untwisted-misaligned bilayer graphene, UMBLG), where the green arrow represents the displacement $D_{m}$.

**Table 1 ijms-25-12549-t001:** Geometrical characteristics of UMBLG associated with TBLG configuration. $D_{m}$ is the displacement in a UMBLG configuration associated with commensurate angle [46] with $\theta_{m}$ (TBLG), $\lambda_{m}$ is the corresponding spatial period of Moiré pattern [47], and *n* is approximately the lattice constant *a* number such that $\lambda_{m}\approx na$.

$D_{m}$ (nm)	$\theta_{m}$ (**°**)	*m*	*n*	$\lambda_{m}/2$ (nm)
$D_{33}=0.122948$	0.987	33	58	7.13718
$D_{32}=0.123024$	1.018	32	56	6.92415
$D_{31}=0.122988$	1.050	31	55	6.71111

## Data Availability

The data presented in this study are available on request from the corresponding author.

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
