# Peer review of "The Cooper-Pair Distribution Function of Untwisted-Misaligned Bilayer Graphene"

_ijms, 2024, doi:10.3390/ijms252312549_

Round 1
Reviewer 1 Report
Comments and Suggestions for Authors
Dear Editor,
After reading the paper entitled “Cooper-pairs distribution function of Untwisted-Misaligned
Bilayer Graphene”, it is my opinion that the paper merits to be published. It essentially presents an extension to the study by one of the authors published in Diamond and Related Materials around 2019, Ref. 15 - https://doi.org/10.1016/j.diamond.2019.04.004. This extension is worth considering given the recent evidence of superconductivity reported on untwisted graphene. Nonetheless, aside minor observations that can be dealt before the proof-reading stage, the only major concern I have with this manuscript is its apparent lack of connection with the most general audience in IJMS, which van be solved but demands some further work. The authors must remember that IJMS which is a forum for biochemistry, molecular and cell biology, molecular biophysics, molecular medicine, and all aspects of molecular research in chemistry rather than on physics, so this apparent disconnection between the scope of the journal and the more physics-driven content of the paper may hinder the impact of this publication. It will be up to the editor to decide whether the paper fits within the scope of the journal or not, despite its good quality.
The aspects that I would suggest to the authors to improve before publication are:
1. Elaborate at the introduction level a clearer connection between this topic and the scope of the journal.
2. Delete any comment of the sort of “calculated and analized for the first time” as if it is the contrary the manuscript will not be publishable.
3. Reconsider the wording used within the first 10 lines of the second paragraph in the introduction. I particularly disagree with the first sentence, as that is just a speculation, but more importantly, the overall paragraph that leads the scope of the paper assumes in advance that the relevant mechanism for the formation of Cooper Pairs in bilayer graphene structures, twisted or untwisted, if phononic. This is at best an hypothesis that allows the treatment of the problem by using Eliashberg theory, but it is no other than an hypothesis and should be defined as such, mentioning as well the other candidates of pairing in competition. For instance, this is relevant to the case here studied because it is already known that the formation of a Moiré pattern in the studied structure could give rise to charge density waves mediating the pairing of electrons. Likewise, this can also enhance the formation of excitons, which could then mediate also the pairing of electrons.
4. Explain in greater detail the similarities or differences in the results obtained at Figs. 3 & 4 of the submitted manuscript and those in the Fig. 2 & 3 of Ref. [15], drawing some conclusion about it regarding the effect of the electric field and the magic angle.
Reviewer 2 Report
Comments and Suggestions for Authors
The paper tackles an intriguing and current topic by investigating the superconducting behavior of Untwisted-Misaligned Bilayer Graphene (UMBLG) in the presence of an external electric field. This contribution is valuable in the ongoing research on twisted graphene and offers a new insight into how slight displacements and external electric fields can influence superconductivity.
The paper is well-organized, with a clear introduction to the background and motivation, followed by a detailed description of methods and results. However, the delineation between sections could be improved to facilitate readability. For instance, discussions on specific results could be more distinctly separated from the general research context in graphene.
The methods used, including ab initio calculations and the use of Cooper-pair distribution functions, are appropriate for the study's goals. However, greater transparency in describing specific algorithms and computational parameter settings would enhance the reproducibility of the study by other researchers.
Data analysis is thorough, and interpretations appear to be supported by the results obtained. Nonetheless, this could be enhanced by a more extensive comparison with other similar studies and by discussing the potential limitations of the methods used.
The conclusions are well-tied to the study's objectives and findings, providing a useful synthesis of discoveries and suggestions for future research. A more detailed discussion of the implications of these findings within the existing theoretical framework and potential practical applications would be beneficial.
The article is very interesting.
Author Response
Thank you very much for taking the time to review this manuscript.
The final version of the manuscript is attached, in which all the adjustments made have been highlighted.
Thank you for your consideration of our work,
Yours,
The Authors (José Camargo, Ivan Gonzalez, and Fredy Mesa).

Reviewer 3 Report
Comments and Suggestions for Authors
The authors study, through the Cooper-pair distribution function, the superductivity in untwisted-misaligned bilayer graphene; they find superconductivity for some values of the parameters. Since twisted bilayer graphene may contain regions of this kind, they propose to exploit these results also to explain the superconductity which has been measured, for certain angles, in twisted bilayer graphene.
The paper reports new results that could be useful to understand the experimentally observed results. However, I think that it needs several improvements in the English form and a few clarifications about the adopted approach and the reported conclusions. Therefore, I suggest the following revisions.
1) The Authors state that the superconductivity observed in twisted bilayer graphene may originate from small regions. Maybe I am wrong, but in order this to be true, isn't it at least necessary for these regions to be sufficiently wide and interconnected? May you clarify better this point?
2) In the introduction I suggest to add a few general references on graphene, such as:
Geim, A. K.; Novoselov, K. S. The rise of graphene. Nat. Materials 2007, 6, 183–191, doi: 10.1038/nmat1849 ;
Castro Neto, A. H.; Guinea, F.; Peres, N. M. R.; Novoselov, K. S.; Geim, A. K. The electronic properties of graphene. Rev. Mod. Phys. 2009, 81, 109, doi: 10.1103/RevModPhys.81.109 ;
McCann, E.; Fal'ko, V. I. Landau-Level Degeneracy and Quantum Hall Effect in a Graphite Bilayer. Phys. Rev. Lett. 2006, 96, 086805, doi: 10.1103/PhysRevLett.96.086805 ;
Marconcini, P.; Macucci, M. Transport Simulation of Graphene Devices with a Generic Potential in the Presence of an Orthogonal Magnetic Field. Nanomaterials 2022, 12, 1087, doi: 10.3390/nano12071087
3) Whenever it appears, substitute "Couper-pairs distribution function" with "Couper-pair distribution function" (without "s", since it has an adjectival role); morever, please add "the" before "Couper-pair distribution function", "$D_{cp} (\omega, T_c)$ function", "$N_{cp}$ parameters" whenever it is needed (i.e., nearly all the times it appears)
4) Abstract, line 6: "consistency" -> "consistent"
5) Introduction, line 3: "properies;" -> "properies:", "suppressing" -> "suppression of"
6) Introduction, line 9: "moir\'e" -> "Moir\'e"
7) Introduction, line 13: "These quite specific structures drive towards providing needed characteristics to the" -> "These quite specific structures may provide the characteristics which are needed in order to make the"
8) Introduction, line 16: "by" -> "through"
9) Introduction, line 19: "corresponded with" -> "characterized by"
10) Introduction, line 20: "are performed" -> "is performed"
11) Introduction, lines 20-22: "it turns to end a ...function" -> "a ... function is obtained"
12) page 2, line 8: "superconducting" -> superconduction"
13) section 2, line 7: explain better what is "$\alpha^2 F(\omega)$"
14) section 2, lines 7-8: "perperpendicular electric fields;" -> "the perperpendicular electric fields:"
15) section 2, line 11: "the phononic" -> "for the phononic"
16) in the two lines after Eq. (1) the Authors should explain better the meaning of the different quantities $g$ which appear here
17) first line after Eq. (2): "Where" -> "where"
18) second line after Fig. 1: "at a small" -> "by a small"
19) first line after Eq. (3): "allows" -> "which allows"
20) line 5 after Eq. (3): clarify exactly the meaning of "are situated particularly" (for example: "are symmetric" or something more proper)
21) line 11 after Eq. (3): "of three" -> "of the three"
22) line 15 after Eq. (3): "a of 12" -> "a 12"
23) caption of Fig. 2, second line: "long-wavelength" -> "a long-wavelength"; "inset of" -> "inset of the"
24) caption of Table 1, last line: write "a" in italics
25) section 3, line 1: "constant of" -> "constant"
26) section 3, line 3: "to AB" -> "for AB"; "due to" -> "due to the fact that"
27) section 3, line 19: "insensitive" -> "insensitivity"
28) caption of Fig. 3: "at $D_{33}$" -> "for $D_{33}$"
29) caption of Fig. 4: "No-twisted" -> "a Not-twisted"
30) line 3 after Fig. 4: "MUBLG" -> "UMBLG"
31) line 5 after Fig. 4: ", it is" -> "is"
32) line 8 after Fig. 4: "rotation small" -> "small rotation"
33) line 11 after Fig. 4: "little misaligned" -> "slightly misaligned"
34) line 13 after Fig. 4: "b)" -> "b))"
35) page 6, line 3: "MUBLG" -> "UMBLG", "gaussian" -> "Gaussian"
36) 2 lines before Fig. 5: "with two" -> "with the two"
37) caption of Fig. 5: "at $D_{33}$" -> "for $D_{33}$"; "no misaligned" -> "for not-misaligned"
38) line 2 after Fig. 5: "obtained of" -> "obtained for"
39) line 7 after Fig. 5: "calculations to" -> "calculations for"
40) line 8 after Fig. 5: "ragarding" -> "corresponding"
41) page 7, line 6: "D_{cp}(\omega,c)" -> "D_{cp}(\omega,T_c)", "due to" -> "due to the fact that"
Comments on the Quality of English Language
The paper needs several improvements in the English form, some of which I have suggested in the previous form.
